# Correlation of single-round strokes & performance indicators: Paris Olympic elite female golfers

Heng Liu[1], Zhenjun Li[2]*, Hongyu Zhou[3], Yingzhong Xie[1], Baohua Liu[3]*, Huang Lin[3]

**1** School of Sports Training, Tianjin University of Sport, Tianjin, China, **2** School of Tourism and Public Administration, Zhuhai College of Science and Technology, Zhuhai, China, **3** School of Social Sports, Tianjin University of Sport, Tianjin, China

* 635962984@qq.com (ZL); liubaohua8259@163.com (BL)

## Abstract

Despite golf's Olympic reinstatement, quantitative research on elite female Olympic golfers remains scarce, with most studies focusing on tour or amateur players. This study aimed to quantify relationships between single-round strokes and 13 performance indicators among top 10 2024 Paris Olympic female golfers (including ties), identify key predictors, and build a predictive model using correlation analysis, principal component analysis (PCA) and LASSO regression with leave-one-out cross-validation (LOOCV) and bootstrap validation to address overfitting concerns inherent to small-sample analyses. Results showed single-round strokes correlated negatively with GIR ($r = -0.484$), SG: Putting ($r = -0.567$) (both $p < 0.01$) and positively with Putt-Total ($r = 0.582$, $p < 0.01$). PCA extracted five principal components explaining 77.94% of total variance, representing distinct performance constructs. LASSO regression identified seven stable predictors (bootstrap selection frequency ≥ 90%): Greens in Regulation (100%), Putt-Total (99%), SG: Approach the Green (98%), SG: Around the Green (97%), SG: Putting (94%), SG: Off the Tee (92%), and Scrambling (88%). Model performance yielded an adjusted R² of 0.928 with cross-validated MSE of 0.528 based on 48 rounds from 12 golfers, though external validation is warranted. This study fills Olympic golf research gaps, providing evidence for targeted training and tournament strategy optimization.

## Introduction

Golf made its Olympic debut at the 1900 Paris Olympics but was discontinued after the 1904 St. Louis Olympics. It was not until the 2016 Rio de Janeiro Olympics that the sport was officially reinstated, marking a pivotal milestone in its global integration [1,2]. For elite female golfers, the Olympics represent the pinnacle convergence of amateur and professional competition, with qualification determined by the Rolex Women's World Golf Ranking (WWGR) overseen by the International Golf Federation

**Data availability statement:** All relevant data are within the paper and its Supporting information files.

**Funding:** The author(s) received no specific funding for this work.

**Competing interests:** The authors have declared that no competing interests exist.

(IGF). The qualification system for the 2024 Paris Olympics requires players to rank within the top 60 of the WWGR (as of June 17, 2024). Each country may send a maximum of four players if all qualify within the top 15; otherwise, the limit is two players. This system ensured that only the most consistent and highest-performing players who had distinguished themselves on the LPGA Tour and in other global competitions qualify for the Paris Olympics, making the field a true representation of the "cream of the crop" in female's golf [3,4]. The evolution of WWGR itself reflects the complexity of athletic performance, weighting tournament strength (with higher coefficients for major championships like the U.S. Women's Open) and incorporating a two-year rolling score to reward sustained performance rather than short-term peaks. This demonstrates that competitive level assessments increasingly rely on quantifiable performance indicators rather than anecdotal skill evaluations [5,6]. For Olympic athletes, qualifying through the WWGR not only validates their technical proficiency but also highlights their ability to adapt to diverse course conditions, competitive pressure, and fatigue [7–9]. Le Golf National, the venue for the 2024 Paris Olympic golf tournament, posed a uniquely rigorous challenge for elite female golfers, characterized by significantly narrower fairways (25−28 yards vs. the LPGA Tour average of 33−36 yards) and rough area maintained at 3.5 inches substantially higher than the tour standard of 2.5 inches. This course configuration, combined with faster green speeds, prioritized driving accuracy and strategic course management, distinguishing the Olympic competition context from conventional LPGA Tour events. In the realm of golf, capturing quantitative metrics of swing mechanics, shot accuracy, and course strategy has become an indispensable tool for players, coaches, and analysts. It is critical to emphasize that the scientific application of performance indicators hinges on their construct validity (ability to accurately measure the intended technical or strategic constructs) and reliability (consistency of measurements across contexts), as is standard in sport-specific assessment development [10]. Traditional performance indicators such as greens in regulation, average putts per round, and driving accuracy have long been used to diagnose weaknesses. LPGA professional players with greens in regulation percentage below 65% will face significantly increased difficulty in advancing to the moving day of tournaments [11]. James (2009) emphasized that performance indicators were not mere "scorecards" but rather "performance blueprints": elite golfers use them to optimize training content (e.g., prioritizing short game practice if sand scrambling fall below 50%) and adjust tournament strategies (e.g., selecting a 7-iron over an 8-iron if launch angle data indicates greater consistency with the former) [12]. For female golfers, their winning margins are typically narrower, making performance indicators even more critical: a 1% increase in putting success rate translates to a 0.3-stroke reduction per round, which could turn a runner-up into a champion [11,13]. Elite female golfers exhibit distinct performance patterns that differ from lower-ranked players and also diverge from male professional golfers. Research indicates that LPGA players' greens in regulation and putting performance account for 60%−70% of single-round strokes variation, significantly higher than driving distance (15%−20%) [12,14]. Another key factor is sensitivity to fatigue and course conditions. At the French National Golf Course, with the Women's Individual Stroke Play pars of 72, The course

spans 6,374 yards, meaning players walk at least 6 kilometers per round while executing repetitive swings. Dehydration or glycogen depletion can reduce putting accuracy by 10%−15%, which is particularly critical for single-round performance [15]. The Olympic format features a four-round stroke play competition, demanding consistent play over consecutive days. Signle round strokes often determine advancement and medal outcomes. Despite the growing emphasis on data within the golfing world, significant gaps remain in research on elite female Olympic golfers. Existing research has primarily focused on LPGA/PGA Tour data or amateur players [3,8,11,16], with very little research conducted on Olympic games [2]. The reason is that the 2016 and 2020 Olympics generated limited performance data, lacking more precise data reporting. Furthermore, past Olympic data analysis has primarily been descriptive rather than quantitative research examining the correlation between performance indicators and single-round strokes. The Olympic course was designed for fairness and spectator appeal, differing from tour courses in fairway length, rough height, and green speed [9]. This means performance indicators observed on the LPGA Tour may not correlate with strokes in the Olympic setting. Previous studies often relied on bivariate correlation analysis without accounting for multicollinearity among performance indicators, potentially limiting understanding of which indicators independently drive single-round performance. Few studies have employed dimensionality reduction methods to identify latent constructs aggregating multiple indicators [17,18].

This study aims to quantify the correlation between single-round strokes and 13 performance indicators for the top 10 elite female golfers (including ties) leading up to the 2024 Paris Olympics, identify key indicators driving competitive success, and construct predictive model [19]. Filling the gap in quantitative research on elite women's golf at the Olympics, this study uniquely focuses on the competitive landscape of the Paris Games. Through a multi-method statistical framework, it uncovers the metric correlations underlying female golfers' single-round performance, providing precise training targets for Olympic preparation. Simultaneously, it offers evidence-based insights for golf governing bodies to optimize Olympic course design and refine the technical statistics system for women's tournaments.

## Materials & methods

### Basic information of the golfers

This study analyzed performance data from a total of 12 elite female golfers who finished within the top 10 ranks (including ties) in the Women's Individual Stroke Play at the 2024 Paris Olympics, resulting in 48 rounds (4 rounds per player). To protect participant privacy in accordance with journal policy, all personally identifiable information has been removed. The cohort comprised athletes from diverse geographical regions and physical profiles, representative of the international field at the Olympic Games.

### Procedure

This study analyzed the data from five aspects, as detailed below (see Fig 1a):

Data Quality Inspection: Completeness check: Ensured no missing values existed across all indicators; samples with missing values were excluded from the analysis. Outliers in the data set were identified and handled appropriately (e.g., deletion, replacement, or separate analysis). Accuracy check: Verified data sources and collection methods to ensure accuracy; data validation and cross-checking were conducted to improve data reliability. Consistency check: Confirmed the uniformity of data units and formats.

Correlation Analysis: Correlation coefficients between indicators were calculated to reflect the strength of linear relationships. The correlation between indicators was determined based on the magnitude of correlation coefficients and significance levels. A heat-map of the correlation matrix was plotted to visually observe the correlations among indicators.

Principal Component Analysis (PCA): As a dimensionality reduction method, PCA transformed multiple correlated indicators into a small number of uncorrelated comprehensive indicators. Based on the interpretation of these principal

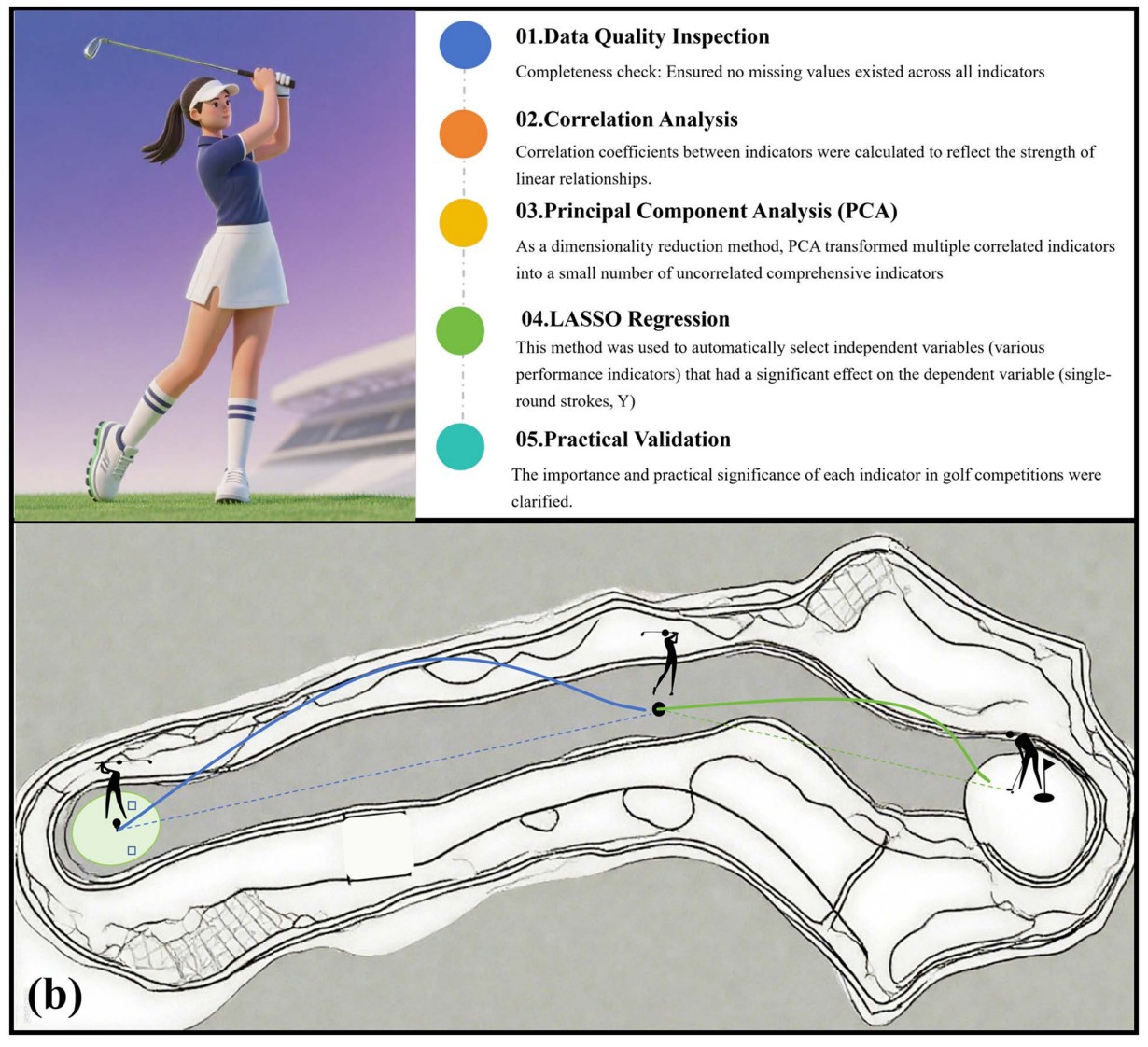

**Fig 1. Technical route and stroking demonstration. (a)** Writing a technical road-map; **(b)** A hole stroke demonstration.

components, those exerting significant impacts on golfers' competitive performance were selected, and their associated indicators were retained.

LASSO (Least Absolute Shrinkage and Selection Operator) regression was employed for variable selection and model construction. Given the relatively small sample size (n = 48) and the number of predictors (p = 13), leave-one-out cross-validation (LOOCV) was used to determine the optimal regularization parameter (λ). The λ value within one standard error of the minimum cross-validation error (λ.1SE) was selected to obtain a more parsimonious model. Bootstrap resampling (n = 100 iterations) was performed to assess the stability of variable selection. Variables with selection frequency ≥ 70% were considered highly stable. All predictor variables were standardized (z-score) prior to analysis. Statistical analyses were performed using MATLAB R2021a (MathWorks, Natick, MA, USA).

Practical Validation: The importance and practical significance of each indicator in golf competitions were clarified. Key indicators were identified based on the purpose of data screening and practical application scenarios.

## Concept definitions

Single-Round Strokes ($Y$): The number of strokes a player uses in a round.

Driving Distance ($X1$/m): Average distance from tee to stopping point during the round (limited to Par4 and Par5 holes).

Driving Accuracy ($X2$/%): The ratio of the number of times the ball comes to rest on the fairway after being struck from the tee area during a round to the total number of Par4 and Par5 holes.

SG (Strokes Gained): Off the Tee ($X3$): The number of strokes a player takes from a specific distance off the tee on par 4 and par 5 is measured against a statistical baseline to determine the player's strokes gained or lost off the tee on a hole. The sum of the values for all holes played in a round minus the field average strokes gained/lost for the round is the player's Strokes gained/lost for that round. The sum of strokes gained for each round are divided by total rounds played.

Greens in Regulation (GIR) ($X4$): The number of strokes taken from the tee to the green in regulation during a round (for Par 3, 4, and 5 holes, the strokes required to reach the green in regulation are 1, 2, and 3, respectively).

Approach Shot Distance to Pin ($X5$/m): The average distance the ball comes to rest from the hole after the player's approach shot.

SG: Approach the Green ($X6$): The number of Approach the Green strokes a player takes from specific locations and distances are measured against a statistical baseline to determine the player's strokes gained or lost on a hole. The sum of the values for all holes played in a round minus the field average strokes gained/lost for the round is the player's Strokes gained/lost for that round. The sum of strokes gained for each round are divided by total rounds played.

Chip Shots-Distance to Pin ($X7$/m): The average distance from the hole after shots within 27.43m (30yards) of the green during a round.

SG: Around the Green ($X8$): The number of Around the Green strokes a player takes from specific locations and distances are measured against a statistical baseline to determine the player's strokes gained or lost on a hole. The sum of the values for all holes played in a round minus the field average strokes gained/lost for the round is the player's Strokes gained/lost for that round. The sum of strokes gained for each round are divided by total rounds played.

Scrambling ($X9$/%): The ratio of the number of times a player made par or better on a par-unreachable green in regulation to the total number of par-unreachable greens in regulation during a round.

Putt-Total ($X10$): The total number of putts used in a round of play.

One putt in Hole ($X11$): The total number of holes in a round where only one putt was used per hole.

One putt in Hole-Distance ($X12$/m): The average distance covered by a single putt on a hole during a round of play.

SG: Putting ($X13$): The number of putts a player takes from a specific distance is measured against a statistical baseline to determine the player's strokes gained or lost on a hole. The sum of the values for all holes played in a round minus the field average strokes gained/lost for the round is the player's strokes gained/lost for that round. The sum of strokes gained for each round is divided by total rounds played.

## Statistical analysis

All competition data were sourced from the official Olympic website (https://olympics.com/). All raw data were in Supplementary Material 1. Microsoft Excel 2021 (Microsoft Corporation, Redmond, WA, USA) was used for data organization

and outlier identification. Statistical analyses were performed using SPSS 27.0 (IBM Corporation, Armonk, NY, USA) and MATLAB R2021a (MathWorks Inc., Natick, MA, USA). LASSO regression was implemented in MATLAB using the lasso function with LOOCV to optimize the regularization parameter (λ). Bootstrap resampling was performed to assess variable selection stability, with predictors selected in ≥ 90% of iterations considered highly stable. Model performance was evaluated using adjusted $R^2$, root mean square error (RMSE), and cross-validated mean squared error (CV-MSE). Statistical significance was set at α = 0.05.

## Results

### Performance indicators

As shown in Table 1, the coefficients of variation (CV) for the Approach Shot Distance to Pin ($X5$), the Scrambling ($X9$), and the One putt in Hole ($X11$) exceed 25%, while the CVs for the remaining performance metrics were relatively low. The CV for the Driving Distance ($X1$) was only 2.41%. Based on the ShotLink golf scoring system, the SG: Off the Tee (X3), the SG: Approach the Green ($X6$), the SG: Around the Green ($X8$), and the SG: Putting ($X13$) represent scoring variables with inherently positive and negative discrete values, making the CV unsuitable for evaluating scoring data.

### Correlation analysis

During competition, players select different clubs to complete the round with the fewest strokes possible; a lower single-round score indicates better performance. As shown in Fig 2, a highly significant negative correlation ($p < 0.01$) was observed between single-round strokes and the following performance indicators: GIR ($r = -0.484$), SG: Approach the Green ($r = -0.445$), Scrambling ($r = -0.476$), One putt in Hole ($r = -0.466$), and SG: Putting ($r = -0.567$). Conversely, a highly significant positive correlation was identified between single-round strokes and the Putt-Total ($r = 0.582$, $p < 0.01$). Additionally, a significant positive correlation was found between single-round strokes and Chip Shots-Distance to Pin ($r = 0.344$, $p < 0.05$), while a significant negative correlation was observed with SG: Around the Green ($r = -0.305$, $p < 0.05$).

**Table 1. Single-round strokes and performance indicators.**

|  | M ± SD | CV (%) |
|---|---|---|
| *Y* | 70.52 ± 2.39 | 3.39 |
| *X1* (m) | 227.08 ± 5.47 | 2.41 |
| *X2* (%) | 64.14 ± 13.32 | 20.77 |
| *X3* | 0.65 ± 1.23 | / |
| *X4* | 12.67 ± 1.80 | 14.17 |
| *X5* (m) | 12.54 ± 6.05 | 48.26 |
| *X6* | 0.97 ± 1.26 | / |
| *X7* (m) | 10.81 ± 1.84 | 14.17 |
| *X8* | 0.26 ± 0.83 | / |
| *X9* (%) | 58.07 ± 20.08 | 34.58 |
| *X10* | 29.35 ± 2.30 | 7.82 |
| *X11* | 6.69 ± 1.89 | 28.33 |
| *X12* (m) | 2.45 ± 0.75 | 30.80 |
| *X13* | 0.62 ± 1.78 | / |

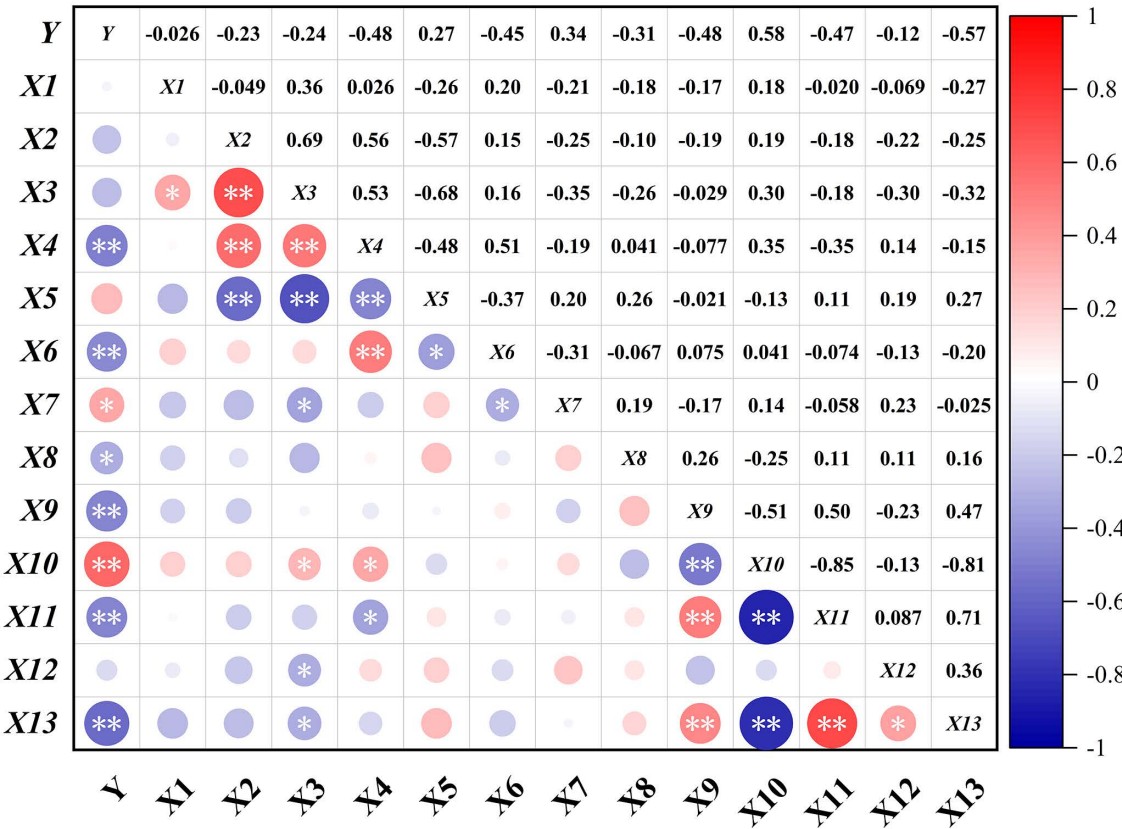

**Fig 2. Heat map of the correlation between the signle round strokes and the performance indicators.** *$p<0.05$, **$p<0.01$.

## Principal component analysis

Principal component analysis (PCA) was performed on the selected performance indicators (see Table 2). Components were screened based on a contribution rate of 70% and eigenvalues≥1. Calculations yielded five principal components: The first principal component (Y1) had an eigenvalue of 4.019 and a contribution rate of 30.918%, reflecting the highest information content; The second principal component (Y2) had an eigenvalue of 2.424 and a contribution rate of 18.645%, reflecting the second-highest information content. The cumulative contribution rate of the third to fifth principal components totaled 28.373%.

Y1 was the first principal component, Y2 was the second principal component, and so on. Table 3 shows that, the first principal component was characterized by positive loadings for SG: Off the Tee (*X3*, 0.186), Putt-Total (*X10*, 0.180), Driving Accuracy (*X2*, 0.163), and GIR (*X4*, 0.157), alongside negative loadings for SG: Putting (*X13*, −0.184) and Approach Shot Distance to Pin (*X5*, −0.162). This component was interpreted as Overall Technical Efficiency, reflecting comprehensive ball-striking and putting performance. The second principal component exhibited strong positive loadings for Scrambling (*X9*, 0.257) and One-putt Percentage (*X11*, 0.244), with negative loadings for Putt-Total (*X10*, −0.249) and Chip Shots Distance to Pin (*X7*, −0.232). This component was designated as the Short Game, representing the ability to save par from difficult positions. The third principal component showed dominant positive loadings for GIR (*X4*, 0.444), One-putt Distance (*X12*, 0.411), and SG: Around the Green (*X8*, 0.322), contrasted with a negative loading for Driving Distance (X1, −0.335). This component was termed the Precision Approach Factor, indicating accuracy-focused rather than distance-focused approach play. The fourth principal component was characterized by positive loadings for One-putt

**Table 2. Eigenvalues and cumulative contribution rates of the indicator correlation matrix.**

| Component | Eigenvalue | Contribution Rate (%) | Cumulative Contribution Rate (%) |
|---|---|---|---|
| Y1 | 4.019 | 30.918 | 30.918 |
| Y2 | 2.424 | 18.645 | 49.563 |
| Y3 | 1.427 | 10.979 | 60.542 |
| Y4 | 1.172 | 9.013 | 69.554 |
| Y5 | 1.090 | 8.381 | 77.935 |
| Y6 | 0.841 | 6.470 | 84.406 |
| Y7 | 0.711 | 5.472 | 89.878 |
| Y8 | 0.537 | 4.134 | 94.012 |
| Y9 | 0.331 | 2.542 | 96.554 |
| Y10 | 0.191 | 1.466 | 98.020 |
| Y11 | 0.126 | 0.970 | 98.990 |
| Y12 | 0.099 | 0.760 | 99.750 |
| Y13 | 0.032 | 0.250 | 100 |

**Table 3. Principal component eigenvalues and feature vectors.**

| Component | Y1 | Y2 | Y3 | Y4 | Y5 |
|---|---|---|---|---|---|
| Eigenvalue | 4.019 | 2.424 | 1.427 | 1.172 | 1.09 |
| Contribution Rate（%） | 30.918 | 18.645 | 10.979 | 9.013 | 8.381 |
| Cumulative Contribution Rate（%） | 30.918 | 49.563 | 60.542 | 69.554 | 77.935 |
| Feature Vector | | | | | |
| *X1* | 0.086 | 0.034 | −0.335 | 0.45 | 0.217 |
| *X2* | 0.163 | 0.145 | 0.203 | −0.165 | −0.384 |
| *X3* | 0.186 | 0.177 | −0.039 | −0.012 | −0.274 |
| *X4* | 0.157 | 0.112 | 0.444 | 0.069 | 0.15 |
| *X5* | −0.162 | −0.195 | −0.03 | −0.066 | 0.138 |
| *X6* | 0.098 | 0.156 | 0.084 | 0.121 | 0.634 |
| *X7* | −0.064 | −0.232 | 0.182 | −0.082 | −0.14 |
| *X8* | −0.086 | 0.005 | 0.322 | −0.334 | 0.310 |
| *X9* | −0.101 | 0.257 | −0.048 | −0.326 | 0.186 |
| *X10* | 0.180 | −0.249 | −0.002 | −0.073 | 0.067 |
| *X11* | −0.162 | 0.244 | −0.091 | 0.141 | −0.131 |
| *X12* | −0.074 | −0.085 | 0.411 | 0.565 | −0.065 |
| *X13* | −0.184 | 0.187 | 0.158 | 0.177 | −0.165 |

Distance (*X12*, 0.565) and Driving Distance (*X1*, 0.450), with negative loadings for SG: Around the Green (X8, −0.334) and Scrambling (*X9*, −0.326). This component was labeled the Distance Advantage Factor. The fifth principal component demonstrated a strong positive loading for SG: Approach the Green (*X6*, 0.634), with a negative loading for Driving Accuracy (*X2*, −0.384). This component was identified as the Approach Shot Performance Factor.

Fig 3 shows that PC1, PC2, PC3, PC4, and PC5 account for 30.9%, 18.6%, 11.0%, 9.0%, and 8.4%, respectively. Only one sample score falls outside the 95% confidence interval, indicating statistical significance.

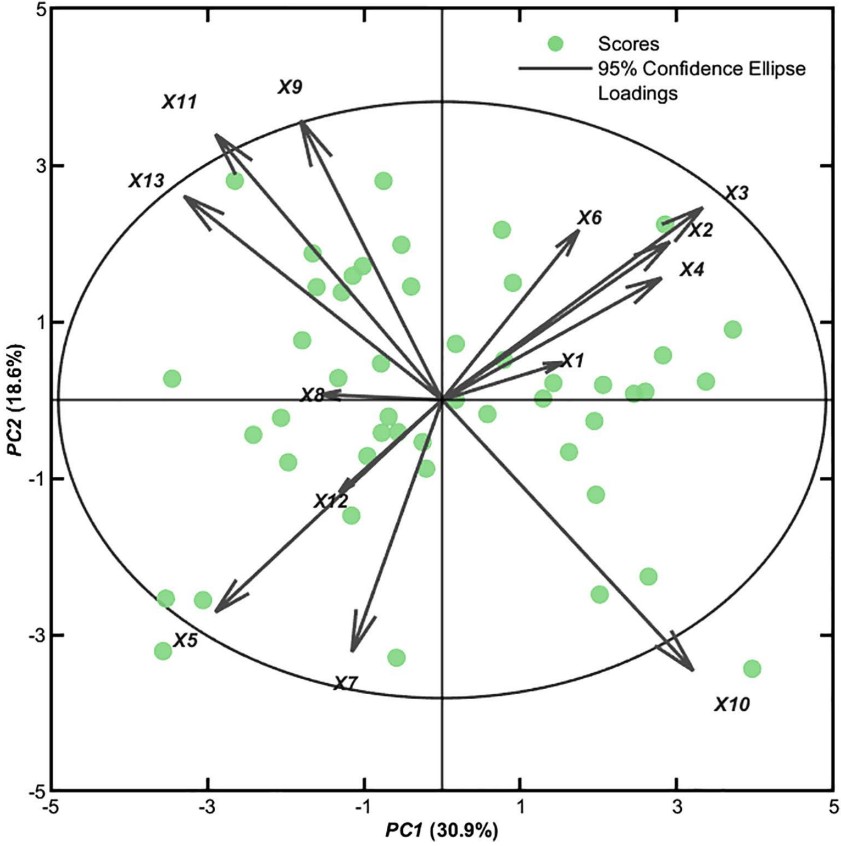

**Fig 3. The players of Biplot.**

## LASSO regression

LASSO regression with LOOCV identified 9 out of 13 candidate variables as significant predictors (Table 4). The LASSO regularization path and LOOCV results are presented in Fig 4, with the optimal $\lambda = 0.094$ yielding minimum cross-validated MSE. Bootstrap validation (n = 100 iterations) confirmed the stability of variable selection. Seven variables demonstrated high stability (selection frequency ≥ 90%): *X4* (100%), *X10* (99%), *X6* (98%), *X8* (97%), *X13* (94%), *X3* (92%), and *X9* (88%). The strongest positive predictor was *X10* ($\beta = 1.257$, 95% CI: 0.548–1.722), followed by *X7* ($\beta = 0.054$, 95% CI: 0–0.344). The strongest negative predictors were *X4* ($\beta = -1.159$, 95% CI: −1.678 to −0.628) and *X13* ($\beta = -0.562$, 95% CI: −1.079 to 0). The model demonstrated good fit (adjusted $R^2 = 0.928$, RMSE = 0.576) with cross-validated MSE of 0.528.

## Discussion

This study systematically analyzed the relationship between single-round strokes and 13 performance indicators among elite female golfers at the 2024 Paris Olympics using correlation analysis, PCA, and LASSO regression. Recent research has demonstrated that biomarker responses can differ significantly between competitive match conditions and standardized testing environments, even within the same athletes [20]. In the unique environment, that is, the data collected in the Olympic Games, it cannot be directly applied to the tour, but it is still of reference significance for the national team's training camp and preparation for the Olympics. On the competition course, the size of the greens ranged approximately

**Table 4. LASSO regression results with bootstrap validation.**

| Variable | β (1SE) | 95% CI | Selection Freq (%) | Stability |
|---|---|---|---|---|
| X10 | 1.257 | [0.548, 1.722] | 99 | High |
| X4 | −1.159 | [-1.678, -0.628] | 100 | High |
| X13 | −0.562 | [-1.079, 0] | 94 | High |
| X6 | −0.453 | [-0.807, -0.001] | 98 | High |
| X3 | −0.373 | [-0.724, 0] | 92 | High |
| X8 | −0.322 | [-0.693, 0] | 97 | High |
| X1 | −0.164 | [-0.304, 0] | 81 | Moderate |
| X9 | −0.154 | [-0.434, 0] | 88 | High |
| X7 | 0.054 | [0, 0.344] | 69 | Moderate |

Note: β, standardized regression coefficient; CI, bootstrap confidence interval. Stability: High (≥ 90%), Moderate (70–89%). Variables X2, X5, X11, X12 were excluded (selection frequency < 30%).

from 300 to 600 m², with each green varying in shape and area (see Fig 1b). Once a player's approach shot had landed on the green defined as any contact with the putting surface resulting Approach Shot Distance to Pin exhibited a considerable standard deviation [21,22]. Consistent with the findings of Brožka et al. (2023), this study indicates that short-game techniques and putting-related metrics were more susceptible to internal and external factors than full-swing indicators [14]. Furthermore, players encountered varying putting lines and distances on the greens, which led to fluctuations in the number of One putt in Hole. The differing rates of GIR achievement among players were identified as a major contributing factor to the high coefficient of variation observed in scrambling success rates [23]. A higher number of GIR and greater SG: Approach the Green per round created advantageous conditions for generating SG: Putting, while holing more one-putts was associated with lower overall strokes. SG indicators quantify deviations in player performance relative to tour averages. Their significant correlation with single-round stroke counts confirms their validity as performance evaluation tools [9,24]. Additionally, higher SG: Around the Green contributed to improved scrambling success rates, which was identified as a critical factor in maintaining low scores [25]. Conversely, greater proximity to the Chip Shots-Distance to Pin increased the difficulty of putting, making it challenging to maintain a low Putt-Total. This often led to higher strokes and diminished competitive performance [26].

Reaching the GIR was a fundamental predictor of low strokes. This study confirms that higher GIR correlates with lower strokes consistent with previous research [27]. Researchers had found that GIR accounts for 30–40% of scoring variability among PGA Tour players [16]. This study extends that conclusion to elite female Olympic golfers, highlighting its cross-population significance. Researches indicated that putting accuracy was the most critical short game technical factor for high-level golfers, directly reducing the number of strokes taken on the green [28]. This study similarly concludes that higher putting scores can decrease the total number of strokes signle round. Hitting the ball farther from the hole after chipping increases the probability of requiring additional putts. For elite female Olympic golfers, putting accounts for 41.62% of strokes in a single round. Reducing putts directly lowers the total strokes. Correlation matrix analysis indicated that, while controlling for total putts and emphasizing the importance of leaving short putts or even holing out after short-game shots, improving approach shot performance, putting efficiency, and scrambling ability had been essential for players to achieve superior results.

Through principal component analysis, five principal components were successfully identified, providing a new perspective for deepening the understanding of the relationship between single round strokes and various performance indicators. Comprehensive Performance Efficiency Factor, which reflects a player's consistency and accuracy during the tee-off phase while also highlighting their performance on the greens [29]. Driving Accuracy and SG: Off the Tee ensured that players were in advantageous positions when approach the green. Putt-Total and SG: Putting reflected efficiency on the

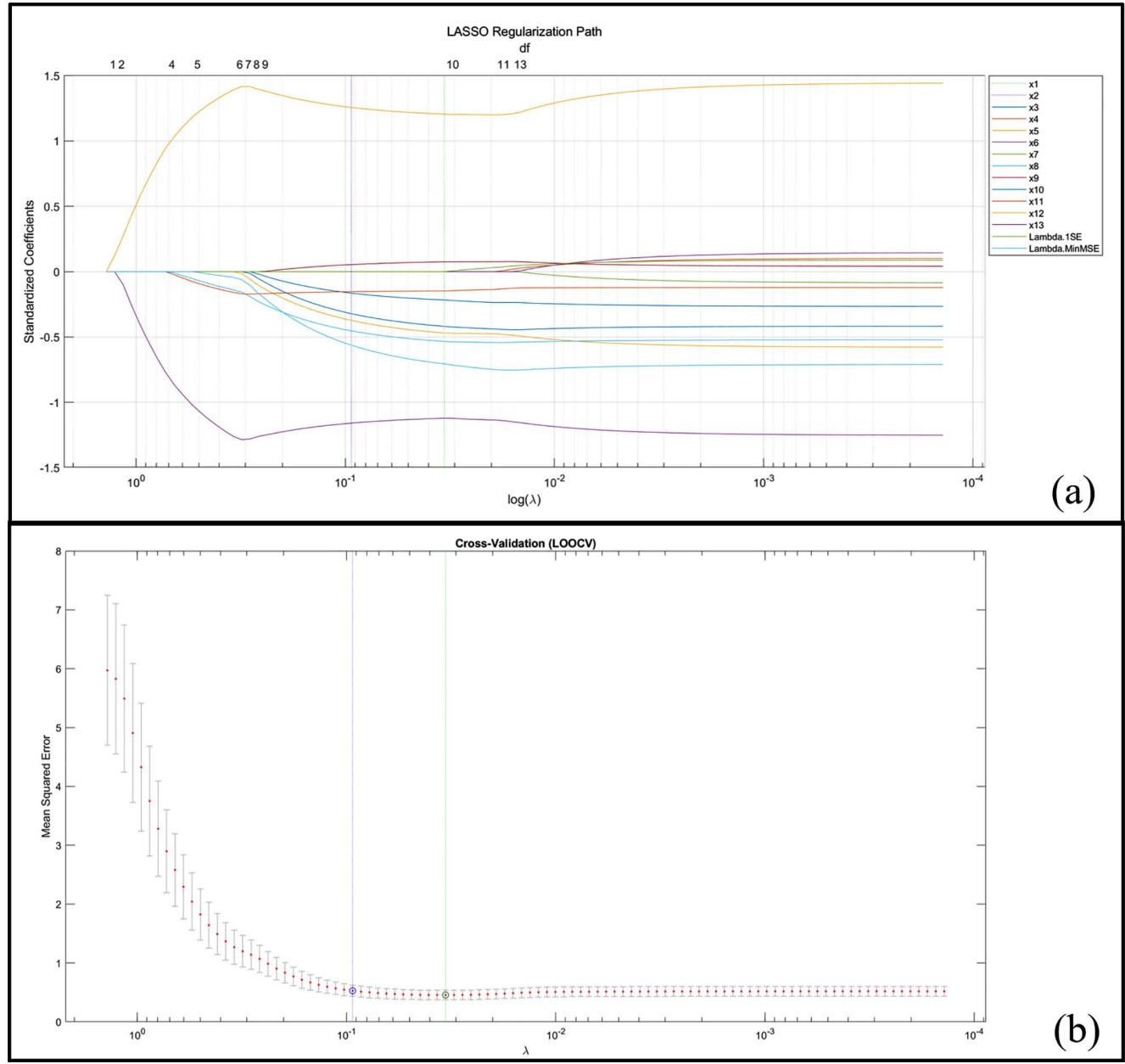

**Fig 4. LASSO regression results. (a)** Regularization path of standardized coefficients. **(b)** Cross-validated mean squared error with leave-one-out cross-validation. Vertical dashed lines indicate Lambda.1SE and Lambda.MinMSE.

green, while distance to hole after approach shot largely determined putting difficulty. Competitive Stability and Adjustment Factor, the signle round strokes reflected overall tournament performance, while Scrambling indicated a player's ability to recover from difficult situations. Chip Shots-Distance to Pin also influenced a player's performance on the greens to a certain extent [30]. Precision Green Performance Factor highlighted a player's accuracy around the putting green. One putt in Hole-Distance reflected a player's ability to sink putts from the green, while GIR indicated putting accuracy. SG: Around the Green further emphasized competitive proficiency in the area surrounding the green [31]. Driving Advantage

Factor reflected a player's distance advantage off the tee. Longer Driving Distance created more opportunities for subsequent shots and reduced the difficulty of approaching the green. However, distance alone was not the sole critical factor; it had to be combined with accuracy and consistency to truly leverage the tee shot advantage [32]. Approach Shot Performance Factor: A high approach shot score indicated a player's ability to land the ball accurately near the pin on the green, creating better conditions for subsequent putting [33]. Through precise shot practice, players improved their GIR rate and SG: Approach the Green; by simulating various challenging scenarios, they enhanced their scrambling success rate; and through dedicated putting training, they increased the one-putt percentage. Additionally, players focused on refining their short game skills to reduce the proximity to the hole after chip and pitch shots [34]. Furthermore, depending on different course layouts and weather conditions, players formulated adaptive competition strategies to leverage their strengths effectively, thereby reducing single-round strokes and achieving better performance.

A critical methodological contribution of this study is the replacement of traditional stepwise regression with LASSO regression, which offers several advantages for our data structure. Given the relatively small sample size (n = 48 rounds) and the number of candidate predictors (p = 13), traditional stepwise regression would likely produce overfitted models with inflated $R^2$ values and unstable variable selection [35]. LASSO regression addresses these concerns through L1 regularization, which simultaneously performs variable selection and coefficient shrinkage, thereby reducing overfitting risk. We employed LOOCV to optimize the regularization parameter, as this approach provides less biased prediction error estimates for small samples compared to k-fold cross-validation [36]. While our model demonstrated strong fit statistics (adjusted $R^2$ = 0.928, CV-MSE = 0.528), these values should be interpreted with appropriate caution. The adjusted $R^2$ represents training set performance and may overestimate predictive accuracy on independent samples. The cross-validated MSE provides a more conservative estimate of model generalizability, though external validation in an independent cohort would be necessary to confirm these findings. Bootstrap validation confirmed that seven of nine selected variables showed high stability (≥90% selection frequency), suggesting that the identified predictors are robust rather than artifacts of overfitting. The findings have several implications for coaching and athlete development in elite women's golf: training programs should prioritize GIR improvement through precision approach practice, as this variable demonstrated the highest selection stability and largest effect size [37]. Second, given that putting accounts for over 40% of strokes, dedicated putting training particularly one-putt conversion from various distances should receive substantial attention [38,39]. Third, short-game recovery skills, including scrambling and around-the-green performance, represent high-value training targets that can compensate for occasional full-swing inaccuracies [40]. Finally, competition strategies should be adapted based on course characteristics and weather conditions to leverage individual strengths effectively [41].

## Limitations and prospects

The most significant limitation is the small sample size (n = 48 rounds from 12 players), which poses a direct threat to the internal validity and stability of our statistical models. Although LASSO regression is specifically designed for scenarios where the number of predictors approaches the sample size [35], and bootstrap validation confirmed reasonable selection stability, the limited sample size restricts our ability to detect subtle effects and may inflate the apparent explanatory power of the model. The events-per-variable ratio (48/13 ≈ 3.7) falls below the commonly recommended threshold of 10–20 for traditional regression, necessitating the use of penalized methods and cautious interpretation. Our findings were derived from a single tournament (2024 Paris Olympics) with unique course characteristics and competitive pressure. Performance relationships observed under Olympic conditions may not generalize to routine tour events or amateur competition. This study analyzed performance indicators without investigating the biomechanical processes that produce these outcomes. Understanding why performance indicators differ between athletes requires analysis of the underlying movement mechanics. Future research should employ wearable technology and AI-driven motion analysis to uncover biomechanical drivers of performance, as these tools are increasingly critical for understanding why certain players achieve superior outcomes [42,43]. Aggregating data across multiple LPGA Tour events would increase sample size and improve model

stability while enabling assessment of context-specific performance relationships. Testing the LASSO model on independent samples from different tournaments would establish the generalizability of the identified predictors.

## Conclusions

This study is the first to quantify performance indicator correlations and key predictors of single-round strokes among 2024 Paris Olympic elite female golfers. Principal component analysis revealed five distinct performance constructs, indicating that elite female golf performance operates through multidimensional technical integration rather than isolated skill dominance. Training programs should prioritize precision approach play to maximize GIR, coupled with systematic putting practice emphasizing distance control and one-putt conversion. Short-game training warrants substantial allocation given its demonstrated role in score preservation under adverse conditions. These results fill gaps in Olympic golf research and provide evidence-based guidance for training, strategy, and course design. Future research should expand the sample and integrate biomechanical/psychological data to further unpack the dynamics of elite female golf performance.

## Supporting information

**S1 File. Raw and processed data.**
(XLSX)

## Acknowledgments

The authors would like to sincerely thank all the authors for their contribution to the completion of this study.

## Author contributions

**Conceptualization:** Heng Liu, Zhenjun Li, Baohua Liu.

**Data curation:** Heng Liu, Zhenjun Li, Hongyu Zhou, Yingzhong Xie, Huang Lin.

**Formal analysis:** Yingzhong Xie, Huang Lin.

**Investigation:** Heng Liu, Zhenjun Li, Huang Lin.

**Methodology:** Hongyu Zhou, Baohua Liu, Huang Lin.

**Project administration:** Heng Liu, Hongyu Zhou.

**Resources:** Zhenjun Li, Yingzhong Xie.

**Software:** Heng Liu, Huang Lin.

**Validation:** Baohua Liu.

**Visualization:** Baohua Liu.

**Writing – original draft:** Heng Liu, Zhenjun Li, Hongyu Zhou, Huang Lin.

**Writing – review & editing:** Heng Liu, Zhenjun Li, Baohua Liu.

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
