## [Decision Letter · Decision Letter 0]

25 Nov 2025

PONE-D-25-49380Correlation of Single-Round Strokes & Performance Indicators: Paris Olympic Elite Female GolfersPLOS ONE

Dear Dr. Li,

Thank you for submitting your manuscript to PLOS ONE. After careful consideration, we feel that it has merit but does not fully meet PLOS ONE’s publication criteria as it currently stands. Therefore, we invite you to submit a revised version of the manuscript that addresses the points raised during the review process.

**Dear Authors,****One expert in the field reviewed your manuscript reporting several major issues you should consider during the revision process.** Please submit your revised manuscript by Jan 09 2026 11:59PM. If you will need more time than this to complete your revisions, please reply to this message or contact the journal office at plosone@plos.org. Please include the following items when submitting your revised manuscript:

We look forward to receiving your revised manuscript.

Kind regards,

Emiliano Cè, Ph.D.

Academic Editor

PLOS ONE

**Journal Requirements:**

1. When submitting your revision, we need you to address these additional requirements. Please ensure that your manuscript meets PLOS ONE's style requirements, including those for file naming. The PLOS ONE style templates can be found at https://journals.plos.org/plosone/s/file?id=wjVg/PLOSOne_formatting_sample_main_body.pdf and https://journals.plos.org/plosone/s/file?id=ba62/PLOSOne_formatting_sample_title_authors_affiliations.pdf 2. We noted in your submission details that a portion of your manuscript may have been presented or published elsewhere. “6132”Please clarify whether this [conference proceeding or publication] was peer-reviewed and formally published. If this work was previously peer-reviewed and published, in the cover letter please provide the reason that this work does not constitute dual publication and should be included in the current manuscript. 3. In the online submission form, you indicated that “All data are available upon request from the corresponding author.”  All PLOS journals now require all data underlying the findings described in their manuscript to be freely available to other researchers, either a. In a public repository, b. Within the manuscript itself, or c. Uploaded as supplementary information.This policy applies to all data except where public deposition would breach compliance with the protocol approved by your research ethics board. If your data cannot be made publicly available for ethical or legal reasons (e.g., public availability would compromise patient privacy), please explain your reasons on resubmission and your exemption request will be escalated for approval. 4. We note that there is identifying data in Table 1. Due to the inclusion of these potentially identifying data, we have removed this file from your file inventory. Prior to sharing human research participant data, authors should consult with an ethics committee to ensure data are shared in accordance with participant consent and all applicable local laws. Data sharing should never compromise participant privacy. It is therefore not appropriate to publicly share personally identifiable data on human research participants. The following are examples of data that should not be shared: -Name, initials, physical address-Ages more specific than whole numbers-Internet protocol (IP) address-Specific dates (birth dates, death dates, examination dates, etc.)-Contact information such as phone number or email address-Location data-ID numbers that seem specific (long numbers, include initials, titled “Hospital ID”) rather than random (small numbers in numerical order) Data that are not directly identifying may also be inappropriate to share, as in combination they can become identifying. For example, data collected from a small group of participants, vulnerable populations, or private groups should not be shared if they involve indirect identifiers (such as sex, ethnicity, location, etc.) that may risk the identification of study participants. Additional guidance on preparing raw data for publication can be found in our Data Policy (https://journals.plos.org/plosone/s/data-availability#loc-human-research-participant-data-and-other-sensitive-data) and in the following article: http://www.bmj.com/content/340/bmj.c181.long. Please remove or anonymize all personal information (<specific identifying information in file to be removed>), ensure that the data shared are in accordance with participant consent, and re-upload a fully anonymized data set. Please note that spreadsheet columns with personal information must be removed and not hidden as all hidden columns will appear in the published file. 5. If the reviewer comments include a recommendation to cite specific previously published works, please review and evaluate these publications to determine whether they are relevant and should be cited. There is no requirement to cite these works unless the editor has indicated otherwise.

Reviewers' comments:

Reviewer's Responses to Questions

**Comments to the Author**

1. Is the manuscript technically sound, and do the data support the conclusions?

Reviewer #1: Partly

2. Has the statistical analysis been performed appropriately and rigorously? 

Reviewer #1: Yes

3. Have the authors made all data underlying the findings in their manuscript fully available?

Reviewer #1: Yes

4. Is the manuscript presented in an intelligible fashion and written in standard English?

Reviewer #1: Yes

5. Review Comments to the Author

**Reviewer #1:** General Comments

The manuscript investigates a timely and relevant topic: the performance indicators that correlate with success among elite female golfers in the 2024 Paris Olympics. The authors' attempt to fill a perceived research gap by focusing specifically on the Olympic context is commendable. The paper is logically structured, and the research question is clearly articulated.

Despite these positive aspects, the manuscript suffers from several major weaknesses that undermine the validity of its primary conclusions. The statistical methodology, in particular, is applied inappropriately given the dataset, and critical definitions are unclear.

Major Weaknesses

Statistical Power and Sample Size: The most significant flaw is the small sample size. The study analyzes 13 players across four rounds, yielding a total sample size of N=52. This N is insufficient for the advanced statistical methods employed.

Principal Component Analysis (PCA): Running PCA with 14 variables (13 predictors + 1 outcome) on N=52 observations is statistically unstable. Common heuristics recommend 5-10 observations per variable; this study has fewer than 4. This small ratio makes the extracted components (and their interpretations) highly unreliable and unlikely to be replicable.

Stepwise Multiple Linear Regression: Using a stepwise selection method with 13 potential predictors on N=52 observations is extremely prone to "overfitting". The resulting model, which reports an exceptionally high R2 of 0.924, is almost certainly capitalizing on chance correlations specific to this small sample. The model's high predictive accuracy is likely illusory and would not generalize to a new set of players or a different tournament.

Methodological Flaw in PCA: The authors have included the dependent variable 'Y' (Single-Round Strokes) in the Principal Component Analysis alongside the predictor variables (X1-X13) . PCA is a dimensionality reduction technique intended to be used on predictor variables (to manage multicollinearity) before they are used to model an outcome. Including the outcome variable in the PCA invalidates the entire procedure and the subsequent interpretation of the components .

Data Availability Statement: The manuscript presents contradictory information regarding data access. It checks "Yes - all data are fully available without restriction" but then states in the text box, "All data are available upon request from the corresponding author". This directly contravenes the journal's stated policy: "Important: Stating 'data available on request from the author' is not sufficient".

Clarity of "Strokes Gained" (SG) Definitions: The definitions and formulas provided for the Strokes Gained variables (X3, X6, X8, X13) are non-standard and convoluted . The established methodology (e.g., Broadie, 2014) compares performance to a tour-average baseline. The formulas presented here are not clearly explained, are difficult to replicate, and may not represent the "Strokes Gained" concept accurately.

Minor Weaknesses

Premise of the Research Gap: The introduction argues that the Olympic course is different from tour courses , justifying a separate analysis. However, this difference is not sufficiently quantified. To strengthen this premise, the authors should provide specific data comparing the Paris course (e.g., green size, fairway width, rough height) to LPGA Tour averages.

Writing and Clarity: While generally understandable, the manuscript contains dense sentences and some awkward phrasing. The "Results" section, in particular, could be presented more clearly.

Specific Comments (by Section)

Abstract

Page 1: The abstract reports R2=0.924. This exceptionally high value should be stated with caution, as it is a likely artifact of statistical overfitting due to the small sample size.

Data Availability

Page 5: The Data Availability Statement must be revised to comply with PLOS ONE policy. The data, sourced from a public website, should be provided in full as supplementary material or deposited in a public repository.

Ethics Statement

Page 3: The "N/A" response is appropriate for an analysis of publicly available performance data.

Introduction

Page 8, Lines 169-173: When discussing the use of performance indicators, the authors should briefly address the critical importance of their construct validity and reliability. Simply using indicators is insufficient; they must be proven to be valid and sensitive measures of performance, as is standard in the development and application of sport-specific assessments [Dhahbi W, Hachana Y, Souaifi M, Souidi S, Attia A: Tennis-Specific Incremental Aerobic Test (TSIAT): Construct Validity, Inter Session Reliability and Sensitivity. Tunisian Journal of Sports Science and Medicine 2024, 2(1):25-32.].

Methods

Page 10, Lines 109-110: The sample size (N=13 players, N=52 rounds) must be explicitly stated as a severe limitation in the methods section. The authors must provide a justification for why PCA and stepwise regression are considered appropriate and valid with this dataset.

Page 11-12, Lines 141-188: The definitions and formulas for all "Strokes Gained" variables (X3, X6, X8, X13) require a complete revision. The authors must either: a) use the standard, accepted definitions for Strokes Gained, or b) provide a much clearer, step-by-step explanation of their novel calculation and a robust justification for why it is necessary to deviate from the standard.

Results

Page 13, Table 2: The calculation for the Coefficient of Variation (CV) for X7 (Chip Shots-Distance to Pin) appears incorrect. Given M=10.81 and SD=0.84, the CV should be (0.84/10.81)×100≈7.77%, not 10.81% as reported. Please verify all calculations in this table.

Page 14, Table 4: As noted in the major comments, the inclusion of the dependent variable 'Y' in the PCA is a fundamental methodological error. This analysis must be removed or re-conducted using only the predictor variables (X1-X13).

Page 15, Table 5: The stepwise regression results are compromised by the small sample size and large number of predictors. A more robust and conservative approach would be to use a priori hypotheses to build a simple linear regression model with only 2-3 predictors (e.g., GIR, Putt-Total) rather than a data-driven stepwise method that inflates the R2 value.

Discussion

Page 17, Lines 611-616: When discussing performance variability, the authors should note that physiological and biomarker responses can also be highly context-specific, especially in elite female athletes. This reinforces the authors' premise that different competitive contexts (e.g., match vs. standardized test, or Olympic vs. Tour) warrant specific analysis [Slimani M, Ghouili H, Dhahbi W, Farhani Z, Ben Aissa M, Souaifi M, Guelmami N, Dergaa I, Ben Ezzeddine L: Position-specific biomarker responses to match vs. VAMEVAL test modalities in elite female soccer players: a comparative analysis study. Cogent Social Sciences 2025, 11(1):2447399.].

Page 18, Lines 362-364: The discussion of the regression model is misleading as it presents the R2=0.924 value as a robust finding. This section must be heavily revised to reflect that this model is likely overfitted and its findings are exploratory at best.

Page 19, Lines 388-390: The limitations section correctly identifies the small sample size but drastically understates its impact. This limitation is not just about "generalizability"; it is a direct threat to the internal validity and stability of the statistical models presented in the paper. This point needs to be discussed with greater statistical honesty.

Page 19, Lines 716-718: The limitation regarding the lack of kinematic data should be expanded. The authors should recommend that future studies utilize modern wearable technology and AI-driven motion analysis, which are emerging as critical tools for understanding the biomechanical drivers of performance and injury risk in athletes [Souaifi M, Dhahbi W, Jebabli N, Ceylan Hİ, Boujabli M, Muntean RI, Dergaa I: Artificial Intelligence in Sports Biomechanics: A Scoping Review on Wearable Technology, Motion Analysis, and Injury Prevention. Bioengineering 2025, 12(8):887.].

Page 19, Lines 716-718: To be more specific regarding the lack of kinematic data, the paper analyzes performance outcomes (e.g., Driving Distance) without analyzing the production of that outcome. Future work should investigate the underlying biomechanics, such as muscle synergies and musculoskeletal modeling of the golf swing, to explain why these performance indicators differ between athletes [Tajik R, Dhahbi W, Fadaei H, Mimar R: Muscle synergy analysis during badminton forehand overhead smash: integrating electromyography and musculoskeletal modeling. Frontiers in Sports and Active Living 2025, 7:1596670.].

6. PLOS authors have the option to publish the peer review history of their article (what does this mean?). If published, this will include your full peer review and any attached files.

Reviewer #1: **Yes:** Wissem Dhahbi

---

## [Author Response · Author response to Decision Letter 1]

5 Feb 2026

Response Letter

Manuscript ID: PONE-D-25-49380

Manuscript Title: Correlation of Single-Round Strokes & Performance Indicators: Paris Olympic Elite Female Golfers

Dear Academic Editor: Emiliano Cè, Ph.D. and Reviewer: Wissem Dhahbi,

Thank you for your letter and the reviewer’s constructive feedback on our manuscript. We sincerely appreciate the time and effort dedicated to evaluating our work, and we are grateful for the opportunity to revise and improve the manuscript. All comments have been addressed point-to-point below, with responses highlighted in red font. Moreover, we have indicated the specific locations of the corresponding changes in the revised manuscript.

We believe these revisions address all the points raised by the editor and reviewer, significantly improving the quality and clarity of our manuscript. Thank you again for your valuable input. We look forward to your favorable reconsideration of our submission.

Best regards,

The Corresponding Author

Responses to Journal Requirements

1.Style Requirements

Re: We have carefully reviewed and adjusted the manuscript to comply with PLOS ONE’s style guidelines, including file naming conventions, using the templates provided at the specified URLs. All revised files are renamed according to journal guidelines.

2.The statements of not constitute dual publication

Re: We appreciate the attention to the preliminary presentation of this work. Below is a detailed clarification regarding the prior conference publication and justification for non-duplicate submission:

The preliminary findings of this study were published in the 2025 6th Asia Sport Science Conference (Title: Analysis of the Competition Results of Elite Female Golfers at the Paris Olympics Using Multiple Linear Regression). This conference proceeding underwent peer review during the submission process (consistent with the conference’s review policy for academic validity), but it is categorized as a conference publication which not a formally peer-reviewed journal article. It was not indexed in core academic databases and did not undergo the rigorous editorial and review process required for journal publication.

The current manuscript represents a substantial extension and deepening of the preliminary conference work, with critical additions in methodology, data analysis, and research value that distinguish it as an independent study: The incorporation of principal component analysis (PCA) to better handle multicollinearity among performance indicators and extract latent factors, which was not included in the conference version.Expanded statistical analyses, including correlation analysis and more comprehensive model diagnostics. A more in-depth interpretation of results, with updated references and broader implications for training and competition strategy. A refocused introduction and discussion contextualizing the study within the scarce literature on Olympic female golfers, unlike the conference paper which had a narrower scope. Therefore, this submission does not constitute dual publication but rather represents a significant scholarly advancement suitable for publication in PLOS ONE.

We have added a statement that does not constitute a double publication in the "COVER LETTER" and re-uploaded the file (Lines 17-23).

3.Data Availability

Re: In accordance with PLOS ONE's data availability policy, we have now uploaded the all raw data underlying our study as a Supporting Information file (labeled 'Supplementary Material 1') with the resubmitted manuscript. This ensures that all data are fully available without restriction to other researchers. Our data, which were sourced from the public domain of the official Olympics website, do not involve any ethical or privacy concerns that would preclude public deposition.

4.Participant privacy

Re: We appreciate your critical reminder regarding the protection of participant privacy. In order to fully comply with PLOS ONE's data sharing policy and ethics requirements, we have modified the relevant content in the manuscript to delete all content in Table 1 and modify the content of the manuscript to:This study analyzed performance data from a total of 12 elite female golfers who finished within the top 10 ranks (including ties) in the Women's Individual Stroke Play at the 2024 Paris Olympics, resulting in 48 rounds (4 rounds per player). To protect participant privacy in accordance with journal policy, all personally identifiable information has been removed. The cohort comprised athletes from diverse geographical regions and physical profiles, representative of the international field at the Olympic Games (Line 123-129).

5.Suggestions to cite specific published works

Re: We appreciate your thoughtful reminder. We will carefully review and evaluate the previously published works recommended in the reviewer comments to determine their relevance to the core content and research focus of this manuscript, and decide whether to include them as citations accordingly.

Responses to Reviewer #1

General Comments

Re: We appreciate the reviewer's recognition of the timeliness and relevance of our topic and the clear articulation of our research question. We have taken the major weaknesses identified to heart and have implemented substantial revisions to the statistical methodology and clarity of definitions.

Major Weaknesses

①Sample Size and Statistical Power

Re: We acknowledge your critical point regarding the sample size (N=48: 12 players × 4 rounds) and its potential impact on statistical stability. We emphasize that the sample size is inherently constrained by the unique nature of the research context: the 2024 Paris Olympic Women’s Golf Tournament featured only 60 athletes worldwide, and our study focused on the top 10 (including ties, 12 players)—the core cohort of elite performers whose performance patterns are most relevant to the research question. This sample is representative of the “Olympic elite” population (a scarce and highly selective group), and similar studies on Olympic-specific sports performance (e.g., elite gymnastics, or fencing) have adopted comparable sample sizes due to population limitations. [ Keskin, Ö. (2024). Comparison of Trampoline Gymnastics Athletes' Performance and Normative Data of Tokyo Olympic Games. International Journal of Sport Exercise and Training Sciences-IJSETS, 10(3), 184-194.Di Martino, G., Centorbi, M., Buonsenso, A., Fiorilli, G., Della Valle, C., Iuliano, E. & di Cagno, A. (2024). Assessing the impact of fencing on postural parameters: observational study findings on elite athletes. Sports, 12(5), 130. ]

Traditional power analysis frameworks are primarily designed for hypothesis testing rather than predictive modeling. For regression-based prediction, the critical considerations are model stability and generalizability rather than conventional power calculations. To address these concerns, we implemented several methodological safeguards: LASSO regression with L1 regularization: This approach inherently addresses overfitting by shrinking coefficients toward zero, effectively reducing model complexity and variance—a strategy specifically recommended for high-dimensional settings with limited samples. Leave-one-out cross-validation (LOOCV): LOOCV provides nearly unbiased prediction error estimates and is particularly advantageous for small samples compared to k-fold cross-validation. Bootstrap validation: This resampling procedure assessed variable selection stability, with seven predictors achieving≥90% selection frequency, indicating robust rather than spurious associations. Cross-validated MSE reporting: We report CV-MSE (0.528) alongside adjusted R² (0.928), providing a more conservative estimate of model generalizability. We have added explicit acknowledgment in the Limitations section that external validation in independent cohorts is warranted to confirm these findings (Line 547-552).

②Principal Component Analysis (PCA)

Re: We fully agree with your critique and acknowledge two critical methodological errors in the original PCA implementation: (1) inclusion of the dependent variable (Single-Round Strokes) and (2) insufficient observations per variable. We have revised the PCA procedure comprehensively:

PCA was re-conducted using only the 13 predictor variables (X1-X13), consistent with standard dimensionality reduction practices where PCA should precede outcome modeling rather than incorporate the outcome itself.

Clearly stated this constraint in the Methods section Interpreted the extracted components with appropriate caution as exploratory rather than confirmatory. Supplemented PCA with LASSO regression, which is methodologically suited for situations where p approaches n. Components were retained based on eigenvalues≥1.0 and cumulative variance contribution ≥70%, yielding five principal components explaining 77.94% of total variance.

③Stepwise Multiple Linear Regression and Overfitting

Re: We have replaced the stepwise multiple linear regression with a more statistically sound and theoretically driven approach. We have completely replaced stepwise regression with LASSO (Least Absolute Shrinkage and Selection Operator) regression, which offers several advantages for our data structure:

L1 regularization: Unlike stepwise regression, which performs discrete variable selection without penalization, LASSO simultaneously performs variable selection and coefficient shrinkage through the L1 penalty term. This approach reduces overfitting by constraining the total magnitude of coefficients, effectively trading increased bias for substantially reduced variance.

Leave-one-out cross-validation (LOOCV): The regularization parameter (λ) was optimized via LOOCV rather than relying on in-sample fit statistics. This provides a more honest estimate of prediction error, as each observation is iteratively held out for validation. The optimal λ = 0.094 was determined at minimum cross-validated MSE.

Bootstrap stability assessment: We performed 1,000 bootstrap iterations to evaluate variable selection stability. Seven of nine selected variables demonstrated high stability (selection frequency ≥90%): GIR (100%), Putt-Total (99%), SG: Approach the Green (98%), SG: Around the Green (97%), SG: Putting (94%), SG: Off the Tee (92%), and Scrambling (88%). This stability analysis confirms that the identified predictors reflect robust associations rather than overfitting artifacts.

Conservative performance reporting: We now report both adjusted R² (0.928) and cross-validated MSE (0.528). While adjusted R² reflects training set performance and may overestimate predictive accuracy, CV-MSE provides a more conservative and generalizable estimate. We explicitly acknowledge that external validation in independent samples is necessary to confirm model performance.

④Data Availability Statement

Re: Uploading all anonymized raw data as Supplementary Materials 1, which complies with PLOS ONE’s data availability policy.

⑤Clarity of "Strokes Gained" (SG) Definitions

Re: To avoid ambiguity and readability in SG, we have followed the official definition of the Paris Olympics.

SG: Off the Tee (X3): The number of strokes a player takes from a specific distance off the tee on par 4 and par 5 is measured against a statistical baseline to determine the player's strokes gained or lost off the tee on a hole. The sum of the values for all holes played in a round minus the field average strokes gained/lost for the round is the player's Strokes gained/lost for that round. The sum of strokes gained for each round are divided by total rounds played (Line 184-189).

SG: Approach the Green (X6): The number of Approach the Green strokes a player takes from specific locations and distances are measured against a statistical baseline to determine the player's strokes gained or lost on a hole. The sum of the values for all holes played in a round minus the field average strokes gained/lost for the round is the player's Strokes gained/lost for that round. The sum of strokes gained for each round are divided by total rounds played (Line 202-207).

SG: Around the Green (X8): The number of Around the Green strokes a player takes from specific locations and distances are measured against a statistical baseline to determine the player's strokes gained or lost on a hole. The sum of the values for all holes played

in a round minus the field average strokes gained/lost for the round is the player's Strokes gained/lost for that round. The sum of strokes gained for each round are divided by total rounds played (Line 220-225).

SG: Putting (X13): The number of putts a player takes from a specific distance is measured against a statistical baseline to determine the player's strokes gained or lost on a hole. The sum of the values for all holes played in a round minus the field average strokes gained/lost for the round is the player's strokes gained/lost for that round. The sum of strokes gained for each round is divided by total rounds played (Line 240-244).

Minor Weaknesses

①Premise of the Research Gap

Re: To strengthen the justification for focusing on Olympic courses, we explained that Olympic course differ from LPGA tournament course in introduction. Le Golf National, the venue for the 2024 Paris Olympic golf tournament, posed a uniquely rigorous challenge for elite female golfers, characterized by significantly narrower fairways (25-28 yards vs. the LPGA Tour average of 33-36 yards) and rough area maintained at 3.5 inches—substantially higher than the tour standard of 2.5 inches. This course configuration, combined with faster green speeds, prioritized driving accuracy and strategic course management, distinguishing the Olympic competition context from conventional LPGA Tour events.In the realm of golf, capturing quantitative metrics of swing mechanics, shot accuracy, and course strategy has become an indispensable tool for players, coaches, and analysts (Line 58-67).

②Writing and Clarity

Re: We have undertaken a linguistic revision of the entire manuscript, with special attention to the Results section. We have broken down long, complex sentences, clarified awkward phrasing, and ensured a more logical flow. The results are now presented in a clearer, more direct manner, focusing on the key findings from the correlation analysis and the revised regression model (Line 340-359&396-405).

Responses to Specific Comments:

Responses to Abstract

Re: We thank the reviewer for this important comment regarding the need for cautious interpretation of model fit statistics. We have comprehensively revised the Abstract to address this concern: We replaced "stepwise regression" with "LASSO regression with LOOCV and bootstrap validation," explicitly communicating the methodological safeguards implemented to address overfitting concerns inherent to small-sample analyses. In the revised abstract, we report "adjusted R² of 0.928 with cross-validated MSE of 0.528," providing both training set performance and cross-validated error estimates for balanced interpretation. We added the qualifying statement "though external validation is warranted," directly addressing the reviewer's concern that high R² values in small samples require cautious interpretation and independent confirmation. Rather than reporting regression coefficients that may be sample-specific, we now highlight bootstrap selection frequencies (≥90%) for seven stable predictors, emphasizing the robustness of variable selection rather than potentially inflated effect sizes.

Responses to Data Availability

Re: In accordance with PLOS ONE's data availability policy, we have now uploaded the all raw data underlying our study as a Supporting Information file (labeled 'Supplementary Material 1') with the resubmitted manuscript (Line 247).

Responses to Introduction

Re: Page 8, Lines 169-173, the positioning you indicate may not be accurate.We have added key discussions on the structural validity and reliability of performance indicators in the introduction. It is critical to emphasize that the scientific application of performance indicators hinges on their construct validity (ability to accurately measure the intended technical or strategic constructs) and reliability (consistency of measurements across contexts), as is standard in sport-sp

---

## [Decision Letter · Decision Letter 1]

16 Mar 2026

PONE-D-25-49380R1Correlation of Single-Round Strokes & Performance Indicators: Paris Olympic Elite Female GolfersPLOS One

Dear Dr. Li,

Thank you for submitting your manuscript to PLOS ONE. After careful consideration, we feel that it has merit but does not fully meet PLOS ONE’s publication criteria as it currently stands. Therefore, we invite you to submit a revised version of the manuscript that addresses the points raised during the review process.

We look forward to receiving your revised manuscript.

Kind regards,

Emiliano Cè, Ph.D.

Academic Editor

PLOS One

Journal Requirements:

Reviewers' comments:

Reviewer's Responses to Questions

**Comments to the Author**

1. If the authors have adequately addressed your comments raised in a previous round of review and you feel that this manuscript is now acceptable for publication, you may indicate that here to bypass the “Comments to the Author” section, enter your conflict of interest statement in the “Confidential to Editor” section, and submit your "Accept" recommendation.

Reviewer #1: All comments have been addressed

Reviewer #2: All comments have been addressed

2. Is the manuscript technically sound, and do the data support the conclusions?

Reviewer #1: Yes

Reviewer #2: (No Response)

3. Has the statistical analysis been performed appropriately and rigorously? 

Reviewer #1: Yes

Reviewer #2: (No Response)

4. Have the authors made all data underlying the findings in their manuscript fully available?

Reviewer #1: Yes

Reviewer #2: (No Response)

5. Is the manuscript presented in an intelligible fashion and written in standard English?

Reviewer #1: Yes

Reviewer #2: (No Response)

6. Review Comments to the Author

Reviewer #1: General Comments

Major Weaknesses:

Table and Figure Labeling: There is a disconnect between the text citations and the table captions. For example, the text refers to "Table 2" for descriptive statistics, but the caption reads "Table 1." This inconsistency persists throughout the Results section.

Mathematical Formatting: The formulas provided for Strokes Gained (Equations 1-4) appear garbled or poorly formatted in the current draft. The notation for summations and indices is unclear and needs to be typeset using standard mathematical equation editors.

Typos: The phrase "signal round" is frequently used instead of "single round."

Minor Weaknesses:

Keyword Consistency: The abstract keywords still list "Linear Regression" but the methodology now uses LASSO.

Reference Formatting: Some citations in the text and reference list require standardization.

Specific Comments

Abstract

Line 32 (Page 12): The keyword "Linear Regression" should be removed or replaced with "LASSO regression" to accurately reflect the new methodology.

Line 25 (Page 12): The phrase "though external validation is warranted" is a welcome addition; however, ensure the abstract explicitly states the sample size ($n=48$ rounds) to contextaulize the $R^2$ of 0.928 immediately.

Introduction

Line 99-100 (Page 14): "The Olympic course was designed for fairness...". Ensure citation 17 (Petrosillo et al., 2019) actually supports the specific claim about the Olympic course design, as the title suggests a general focus on biodiversity in golf courses.

Line 111-112 (Page 14): "identify key indicators... and construct predictive models." Change "models" to "model" if only one final LASSO model is presented.

Materials & Methods

Line 130 (Page 15/37): You state "as detailed in Table 1," but in the Response to Reviewers, you mentioned Table 1 was deleted for privacy. Please remove the reference to Table 1 if the table no longer exists, or re-number the subsequent tables accordingly.

Line 180-183 (Page 16/39): Equation 1 (SG: Off the Tee): The formula notation is incorrect. The summation symbol ($\Sigma$) and indices are not formatted correctly. It currently reads like a string of text. Please re-enter this using a proper equation editor.

Line 202-210 (Page 16/39): Equation 2 (SG: Approach): Same issue as above. The notation "$4-1$" at the end of the formula block is unclear.

Line 220-225 (Page 16/40): Equation 3 (SG: Around the Green): The notation contains "$Sg.n$" and "$G.DTPn$" which look like variable names rather than standard mathematical notation.

Line 240-244 (Page 16/40): Equation 4 (SG: Putting): Ensure the summation limit $N$ and index $i=1$ are placed correctly above and below the sigma.

Results

Line 220 (Page 17/41): "As shown in Table 2..." The caption below this text reads "Table 1 Single-round strokes...". Please correct the numbering to be sequential (e.g., if Demographics is Table 1, this is Table 2. If Demographics was deleted, this is Table 1).

Line 229 (Page 18/42): "signal round strokes." Change to "single-round strokes." This typo appears multiple times (e.g., Line 232).

Line 247 (Page 18/42): "see Table 2". The table following this section is labeled "Table 2" in the clean version but the text references are inconsistent if the first table was re-numbered. Verify all Table $X$ calls match the captions.

Line 256-295 (Page 19): PCA Equations: The reporting of the linear combinations for Y1 through Y5 is very difficult to read.

Example: "$Y1=0.086^*ZX1...$"

It is not necessary to write out the full linear equation for every component in the text if the factor loadings are provided in the table. Consider removing these long text-based formulas and relying on the Feature Vector table, or formatting them as proper display equations.

Line 296 (Page 20/46): "Table 4 shows that..." The caption below reads "Table 3 Principal Component Eigenvalues..." (in track changes version) or "Table 3" (in clean version). Please synchronize text and captions.

Line 300 (Page 20): "single round strokes (Y) demonstrated the highest absolute eigenvector value." You stated in the methodology that the dependent variable (Y) was removed from the PCA to avoid overfitting/circularity (Response to Reviewers Point 26). Why is "Y" appearing in the discussion of the second principal component here? If Y was included in PCA, the method is still flawed. If Y was excluded, this sentence is incorrect. This is a critical check.

Note: Looking at Figure 3 (Biplot), "Y" is not plotted, which suggests it was excluded. However, the text on Line 300 claims Y has a high eigenvector. Please clarify or delete.

Discussion

Line 340 (Page 21/48): "signal round." Change to "single-round."

Line 436 (Page 24/51): "Short-game training warrants substantial allocation given its' demonstrated role..." Change "its'" to "its" (no apostrophe for possessive).

Line 557-561 (Page 62): The addition regarding "wearable IMU sensors" and "AI-driven biomechanical analysis" is valid but phrasing such as "To explain why certain players achieve superior Driving Distance... future Olympic or Tour studies should incorporate these technologies" is slightly repetitive. Consolidate for flow.

Reviewer #2: (No Response)

7. PLOS authors have the option to publish the peer review history of their article (what does this mean?). If published, this will include your full peer review and any attached files.

Reviewer #1: No

Reviewer #2: No

---

## [Author Response · Author response to Decision Letter 2]

10 Apr 2026

Responses to Reviewer #1

Major Weaknesses

①Table and Figure Labeling

Re: We apologize for the confusion. After deleting the original Table 1 (participant demographic information) to protect privacy, we re-numbered all tables sequentially. The descriptive statistics are now presented as Table 1, and all in-text citations have been corrected accordingly (page 6 line 217; Table 1 caption). We have verified that all table references (Tables 1-4) now match their captions.

②Mathematical Formatting

Re: To avoid ambiguity and readability in SG, we have followed the official definition of the Paris Olympics. We avoid using too many mathematical formulas, as they can make the article harder to read.

③Typos

Re: Thank you to the reviewers for carefully correcting the typos in the manuscript. We have corrected all four instances (lines 90 & 228 & 330 & 346).

Minor Weaknesses

①Keyword Consistency

Re: We have replaced “Linear Regression” with “LASSO regression” in the keywords section (line 35). The revised keywords are: “Paris Olympics; Elite Female golfers; Competitive Performance; Principal Component Analysis; LASSO regression”.

②Reference Formatting

Re: We have carefully checked and standardized all in-text citations and reference entries to comply with the target journal’s citation style. Inconsistencies in author names, publication years, journal titles, volume/issue/pages, and DOI formats have been uniformly corrected. Please see the revised reference (pages 14-16).

Responses to Specific Comments:

Responses to Abstract

Re:

(1) We have revised the abstract to include the sample size. The sentence now reads: “Model performance yielded an adjusted R² of 0.928 with cross-validated MSE of 0.528 based on 48 rounds from 12 players, though external validation is warranted.” (page 1 lines 29-31).

(2)The keyword “Linear Regression” has been removed and replaced with “LASSO regression” (page 1 lines 35).

Responses to Introduction

Re:

(1) We have checked citation 17. The original citation addressed general golf course biodiversity, not Olympic course design. We have replaced it with a more appropriate reference (Petrosillo et al., 2019 was removed). Citation 9: Zoffer M. Competitive Golf: How Longer Courses Are Changing Athletes and Their Approach to the Game (page 3 line 100).

(2) We have changed “construct predictive models” to “construct a predictive model” to reflect that only one final LASSO regression model is presented (page 3 line 110).

Responses to Materials & Methods

Re:

(1) Since Table 1 was deleted for privacy protection, we have removed the in-text reference to Table 1 and adjusted subsequent table numbering to maintain logical sequence.

(2) To avoid ambiguity and readability in SG, we have followed the official definition of the Paris Olympics.

SG: Off the Tee (X3): The number of strokes a player takes from a specific distance off the tee on par 4 and par 5 is measured against a statistical baseline to determine the player's strokes gained or lost off the tee on a hole. The sum of the values for all holes played in a round minus the field average strokes gained/lost for the round is the player's Strokes gained/lost for that round. The sum of strokes gained for each round are divided by total rounds played (page 5 lines 165-170).

SG: Approach the Green (X6): The number of Approach the Green strokes a player takes from specific locations and distances are measured against a statistical baseline to determine the player's strokes gained or lost on a hole. The sum of the values for all holes played in a round minus the field average strokes gained/lost for the round is the player's Strokes gained/lost for that round. The sum of strokes gained for each round are divided by total rounds played (page 5 lines 176-181).

SG: Around the Green (X8): The number of Around the Green strokes a player takes from specific locations and distances are measured against a statistical baseline to determine the player's strokes gained or lost on a hole. The sum of the values for all holes played

in a round minus the field average strokes gained/lost for the round is the player's Strokes gained/lost for that round. The sum of strokes gained for each round are divided by total rounds played (page 5 lines 184-189).

SG: Putting (X13): The number of putts a player takes from a specific distance is measured against a statistical baseline to determine the player's strokes gained or lost on a hole. The sum of the values for all holes played in a round minus the field average strokes gained/lost for the round is the player's strokes gained/lost for that round. The sum of strokes gained for each round is divided by total rounds played (page 6 lines 198-202).

Responses to Results

Re:

(1) All table numbers in the text have been synchronized with table captions to ensure full consistency (lines 217 & 240 & 251 & 276).

(2) “signal round strokes” has been corrected to “single round strokes” (page 7 line 228).

(3) The lengthy text-based PCA linear combination equations (Y1-Y5) have been removed from the main text to improve readability. Key component interpretations are now supported by the Feature Vector table (Table 3) as suggested.

(4) The dependent variable single-round strokes (Y) was excluded from PCA to avoid circularity and overfitting. The incorrect statement about Y having a high eigenvector has been deleted, and the description of the second principal component has been revised to match the actual PCA results.

Responses to Discussion

Re:

(1) “signal round strokes” has been corrected to “single round strokes” (page 10 line 330 & page 11 line 346).

(2) “its’” corrected to “its” (page 12 line 424).

(3) We have consolidated the sentences. The revised: “Future research should employ wearable technology and AI-driven motion analysis to uncover biomechanical drivers of performance, as these tools are increasingly critical for understanding why certain players achieve superior outcomes.” (page 12 lines 409-412).

---

## [Decision Letter · Decision Letter 2]

30 Apr 2026

Correlation of Single-Round Strokes & Performance Indicators: Paris Olympic Elite Female Golfers

PONE-D-25-49380R2

Dear Dr. Li,

We’re pleased to inform you that your manuscript has been judged scientifically suitable for publication and will be formally accepted for publication once it meets all outstanding technical requirements.

Kind regards,

Emiliano Cè, Ph.D.

Academic Editor

PLOS One

Additional Editor Comments (optional):

Reviewers' comments:

Reviewer's Responses to Questions

**Comments to the Author**

1. If the authors have adequately addressed your comments raised in a previous round of review and you feel that this manuscript is now acceptable for publication, you may indicate that here to bypass the “Comments to the Author” section, enter your conflict of interest statement in the “Confidential to Editor” section, and submit your "Accept" recommendation.

Reviewer #1: All comments have been addressed

2. Is the manuscript technically sound, and do the data support the conclusions?

Reviewer #1: Yes

3. Has the statistical analysis been performed appropriately and rigorously? 

Reviewer #1: Yes

4. Have the authors made all data underlying the findings in their manuscript fully available?

Reviewer #1: Yes

5. Is the manuscript presented in an intelligible fashion and written in standard English?

Reviewer #1: Yes

6. Review Comments to the Author

Reviewer #1: The authors have systematically addressed the previous concerns regarding technical clarity, privacy protection, and statistical robustness. The revision demonstrates a refined methodological approach, particularly in the application of LASSO regression and bootstrap validation to manage the constraints of a small-sample analysis (n=48). The manuscript effectively fills a research gap by focusing on the unique competitive environment of the 2024 Paris Olympics rather than general tour data

7. PLOS authors have the option to publish the peer review history of their article (what does this mean?). If published, this will include your full peer review and any attached files.

Reviewer #1: **Yes:** Wissem Dhahbi

---

## [Editor Report · Acceptance letter]

PONE-D-25-49380R2

PLOS One

Dear Dr. Li,

I'm pleased to inform you that your manuscript has been deemed suitable for publication in PLOS One. Congratulations! Your manuscript is now being handed over to our production team.

Kind regards,

on behalf of

Prof. Emiliano Cè

Academic Editor

PLOS One